# The Approximate Solution of Nonlinear Flexure of a Cantilever Beam with the Galerkin Method

Jun Zhang [1], Rongxing Wu [1,2], Ji Wang [1,*], Tingfeng Ma [1] and Lihong Wang [3]

[1] Piezoelectric Device Laboratory, School of Mechanical Engineering and Mechanics, Ningbo University, 818 Fenghua Road, Ningbo 315211, China; tiehan99@163.com (J.Z.); wurongxing98@163.com (R.W.); matingfeng@nbu.edu.cn (T.M.)
[2] Institute of Applied Mechanics, Ningbo Polytechnic, 1069 Xinda Road, Beilun District, Ningbo 315800, China
[3] School of Mathematics and Statistics, Ningbo University, 818 Fenghua Road, Ningbo 315211, China; wanglihong@nbu.edu.cn
[*] Correspondence: wangji@nbu.edu.cn; Tel.: +86-574-8760-0467

**Abstract:** For the optimal design and accurate prediction of structural behavior, the nonlinear analysis of large deformation of elastic beams has broad applications in various engineering fields. In this study, the nonlinear equation of flexure of an elastic beam, also known as an elastica, was solved by the Galerkin method for a highly accurate solution. The numerical results showed that the third-order solution of the rotation angle at the free end of the beam is more accurate and efficient in comparison with results of other approximate methods, and is perfectly consistent with the exact solution in elliptic functions. A general procedure with the Galerkin method is demonstrated for efficient solutions of nonlinear differential equations with the potential for adoption and implementation in more applications.

**Keywords:** Galerkin; beam; elastica; nonlinear; flexure; solution

## 1. Introduction

Differential equations are widely used for the description of scientific and engineering phenomena with frequent appearances of nonlinear types [1–4]. Consequently, differential equations and solution methods have a major role in modeling transmission dynamics and rendering control strategies for communicable diseases such as tuberculosis and COVID-19 [5,6]. There have been tremendous efforts to solve differential equations with generalized and ad hoc methods, but there are always problems that cannot be effectively solved for many reasons [7–9]. Some known solutions of some specialized differential equations, such as those related to Bessel functions and hypergeometric functions, can be effectively used, but there are still differential equations that cannot be solved with existing methods and known functions [10,11].

With classical methods of solutions failing in these contexts, in recent decades, some special methods with gradual sophistication have been developed, such as the multiscale method [10], homotopy analysis method [12–15], homotopy perturbation method [16], and others dealing with emerging differential equations in many fields [17,18]. Generally, most of these methods are asymptotic in nature, and can find solutions in series form with an improved converging rate and numerical stability over a relatively large interval. In combination with the algorithm development and computing tools for symbolic computation, these methods can powerfully and efficiently solve many practical and classical nonlinear equations. As a result, many techniques have been applied to the latest problems, and the improved solution techniques have been widely promoted and utilized.

Studies on the large deflection of beams mostly focused on different boundary conditions, cross-section characteristics, and loading conditions [19], among others. Formulated with the angle enclosed by the axes of the deformed and undeformed configurations, the

nonlinear equation of an elastic beam subjected to concentrated forces or couples resembles the equation of motion of a rigid pendulum according to Kirchhoff's kinetic analogy [20]. Large deflections of an end-supported beam subjected to a point load are solved using both the elliptic integral method and the shooting-optimization technique [21]. The homotopy analysis method was used to solve this problem with enough accuracy [13]. In recent attempts to find an optimal method for nonlinear vibrations of structures, it was discovered that the Galerkin method and its alternative form, the Rayleigh-Ritz method, can be effective for approximate analysis of nonlinear vibrations in certain circumstances [22–25]. The Legendre–Galerkin matrix (LGM) method is used to solve the vibration equations of single (SDOF) and multiple degree-of-freedom (MDOF) structural systems [26]. The nonresonance and primary resonance of microbeams with consideration of small-scale effects and nonlinear terms were obtained by the Galerkin and multiple-scale methods [27].

With the objective of finding an effective method and taking the advantage of the widespread availability of powerful symbolic computing tools and resources, it is thought that the Galerkin method can be used for obtaining relatively accurate solutions to nonlinear differential equations with a simple procedure [28,29]. The procedure can be developed with the appropriate trial solutions satisfying essential boundary and initial conditions, making the computing process more efficient for accelerated convergence and improving accuracy [30,31]. Furthermore, if the orthogonality of trial solutions is guaranteed, the calculation will be further optimized. These concepts are already practiced in linear problems with the Galerkin method and its modifications, and further improvements can be made with nonlinear problems as an innovative implementation. The basic ideas outlined here were applied to the nonlinear equation of a thin beam, and the results showed that the analysis can be used, and the results are quite accurate. The demonstration of the use of the Galerkin method for approximate solutions to nonlinear differential equations is novel and enlightening in the promotion of this classical and practical method for new objectives and applications. The robustness of this approach is more apparent when applied to this historical elastica problem, which has been used as a benchmark for various methods and techniques. This application presents an ideal case for employing the Galerkin method to take advantage of its efficiency and simplicity in dealing with increasing nonlinear differential equations from various applications in the absence of analytical or other approximate solutions.

## 2. Galerkin Method for Solutions of Nonlinear Differential Equations

To implement the Galerkin method to solve nonlinear differential equations, the fundamental concepts and procedures are introduced to provide an essential understanding. The rigorous mathematical procedure and associated assumptions and proofs are not included in this paper, but they can be found in many mathematical textbooks and monographs [10,32]. A general nonlinear differential second-order equation is

$$N\left(x, y, y^k, y', y'', p\right) = 0, \ x \in [a, b], \tag{1}$$

where $y, y'(y'')$, $p, x, a$, and $b$ are the function, derivatives with respect to the spatial variable, nonlinear term with the power of $k$, parameter, spatial variable, lower bound, and upper bound, respectively. With prescribed boundary and initial conditions, the approximate solution, also generally asymptotic, can be assumed as

$$y = \sum_{n=0}^{M} A_n Y_n(x), \tag{2}$$

where the chosen function $Y_n(x)$ satisfies the boundary conditions, and $A_n$ are coefficients to be determined with the limitations by an integer $M$. As a general rule of thumb, the trial solutions $Y_n(x)$ can be conveniently chosen from the solutions of a simplified linear equation, so calculations will be simple and efficient. Furthermore, it is important to

expect that these solutions are also orthogonal. There is abundant literature on the optimal selections of trial functions, and one must deal with some typical equation types and appropriate boundary conditions.

Applying the Galerkin method, the problem in Equation (1) is transformed to

$$\int_a^b N\left(x, y, y^k, y', y'', p\right) \sum_{n=0}^M \delta A_n Y_n(x)\, \mathrm{d}x = 0. \tag{3}$$

Because the Galerkin method has proven its capability to provide accurate solutions to linear differential equations as a powerful and robust approximate procedure, it is also thought the nonlinear equations can be solved in a similar manner with equal strength. With this hypothesis in mind, the arbitrary variation in independent coefficients in Equation (3) results in a series of nonlinear algebraic equations of unknown amplitudes after integration as

$$\overline{N}_n\left(A_m,\ A_m^q, p\right) = 0,\ n, m = 1, 2, \cdots, M, \tag{4}$$

using the equation and trial functions with parameters $p$ and $q$. By solving the system of nonlinear algebraic equations for amplitude $A_m$, the approximate solution can be obtained. It is better to have orthogonal functions in the domain of integration for a simple form and least couplings of unknowns in Equation (4). After obtaining solutions of the coefficients, the solution is in Equation (2) for manipulations in relation to the problem.

If a simplified procedure is sought, with a careful selection of trial functions satisfying the boundary conditions, the procedure starts with

$$y = A_0 Y_0(x), \tag{5}$$

Then, the first approximate equation is

$$\int_a^b N\left(x, A_0 Y_0(x), A_0^k Y_0^k(x), A_0 Y_0'(x), A_0 Y_0''(x), p\right) Y_0(x)\mathrm{d}x = 0. \tag{6}$$

The coefficient $A_0$ is to be solved from the above equation. To improve the solution with an additional function, it is further assumed that the updated solution is

$$y = A_0 Y_0(x) + A_1 Y_1(x),\ \ \delta y = \delta A_1 Y_1(x). \tag{7}$$

Then, the following Equation is for the unknown amplitude $A_1$

$$\int_a^b N\left(x, y, y^k, y', y'', p\right) Y_1(x)\mathrm{d}x = 0, \tag{8}$$

by using known $A_0$ from the procedure above in Equation (6). By repeating this procedure with the successive addition of higher-order solutions, other coefficients of higher-order approximations can be successively obtained for an improved solution.

With the two steps for the approximate solution in the above demonstration, the results have to be compared before a conclusion can be drawn on the advantage of the Galerkin method. If a good trial solution is selected from the beginning, the accuracy of solutions will be satisfactory based on the proven trust in the Galerkin method. Further proof and validation can be made with some typical examples. These steps are simple with given equations that are asymptotic in nature. The amplitudes can be improved through an iterative procedure from the nonlinear algebraic equations.

## 3. Equation of Flexure of an Elastic Beam and Its Solution

With the procedure and algorithm outlined above, a typical nonlinear equation is needed to test the procedure and accuracy. Among the many typical nonlinear differential

equations, the large deformation of a beam under a concentrated force at the free end, also known as the elastica, is [14,15,31–33]

$$\theta'' + \alpha \cos \theta = 0, \ \theta(0) = 0, \ \ \theta'(1) = 0, \tag{9}$$

where $\theta = \theta(x)$ is the rotation of the cross-section of beam, and $\alpha = Pl^2/EI$ is a constant of the property of the beam problem, with $P$, $l$, and $EI$ as the point load at the free tip, beam length, and bending stiffness of the beam, respectively.

This is widely known as the elastica problem, which has been extensively studied [32–34]. The elastica of nonprismatic cantilever beams with rectangular cross-sections that are subjected to combined loading was studied by Lee [35]. Wang et al. used the homotopy analysis method (HAM) for this problem with approximate solutions in power series with a large radius of convergence [15]. An elastic beam with relatively large deformation under a concentrated load is a common structural problem in both civil and mechanical engineering [10,11,14,15,32–34]. The analytical methods and exact solution to this problem are very important for the design and optimization of these structures. Therefore, a new solution technique with improved results is important for validating the proposed approach with the Galerkin method. In previous studies, the traditional Galerkin method was first used for analysis of wave propagation in finite solids with nonlinear complications [36]. By assuming the spatial variation in displacements in the exact form of the known linear mode shape, Wang et al. used the orthogonal properties of the linear mode shapes to obtain the equations to approximately determine the temporal behavior [37]. Wang further expanded the application scope of the Galerkin method by integrating the weighted equation of motion over the time of one period of vibrations to eliminate the harmonics from the deformation function [31]. This is an extension of Galerkin method that has broad applications in approximate solutions, particularly for solid mechanics problems [29]. This extended Galerkin method can also be utilized for the analysis of nonlinear differential equations of free and forced nonlinear vibrations of structures as a new technique with many advantages [28–31]. The procedure outlined in this paper for the determination of displacement coefficients from nonlinear algebraic equations is new, but it is a natural extension of the popular Galerkin method meriting wide publicization and promotion.

By examining Equation (9), it is clear that the trial solution with trigonometric functions satisfying the boundary conditions is in the form of

$$\theta(x) = \sum_{n=0}^{\infty} A_n \sin \frac{2n+1}{2} \pi x. \tag{10}$$

The amplitudes $A_n$ are determined with the procedure outlined above. One must substitute Equation (10) into Equation (9) with a standard Galerkin formulation

$$\int_0^1 (\theta'' + \alpha \cos \theta) \delta\theta \mathrm{d}x = 0, \tag{11}$$

where

$$\delta\theta = \sum_{n=0}^{\infty} \delta A_n \sin \frac{2n+1}{2} \pi x, \tag{12}$$

with arbitrary variation in amplitudes $\delta A_n$.

For the evaluation of the integral, there are several choices of analytical expressions and power series expressions. It is more favorable to use

$$\cos \theta = \sum_{n=0}^{\infty} (-1)^n \frac{\theta^{2n}}{(2n)!} = 1 - \frac{\theta^2}{2} + \frac{\theta^4}{24} - \frac{\theta^6}{720} + \ldots. \tag{13}$$

Then, the problem will be the evaluation of individual equations without coupling as

$$\int_0^1 (\theta'' + \alpha \cos \theta) \sin \frac{2n+1}{2}\pi x \, dx = 0, \quad n = 1, 2, 3, \cdots, M. \tag{14}$$

With Equation (4), Equation (6) can be written as

$$\int_0^1 \left[ -\sum_{n=0}^N \left(\frac{2n+1}{2}\pi\right)^2 A_n \sin \frac{2n+1}{2}\pi x \right. $$
$$\left. +\alpha \cos \sum_{n=0}^N A_n \sin \frac{2n+1}{2}\pi x \right] \sin \frac{2n+1}{2}\pi x \, dx = 0 \tag{15}$$
$$n = 1, 2, 3, \cdots, M,$$

for the coupled amplitudes with the choice of order $M$. The procedure outlined above is followed to obtain the solutions of amplitudes.

To obtain the amplitude $A_0$, the approximate solution is taken as

$$\theta = A_0 \sin \frac{\pi}{2}x, \tag{16}$$

and the integration of Equation (7) yields

$$575 - \frac{1575\pi^3 A_0}{16\alpha} - 525A_0^2 + 35A_0^4 - A_0^6 = 0. \tag{17}$$

To find appropriate solution of $A_0(\alpha)$, the curves from Equation (17) are plotted in Figure 1 to help the solution procedure of this equation.

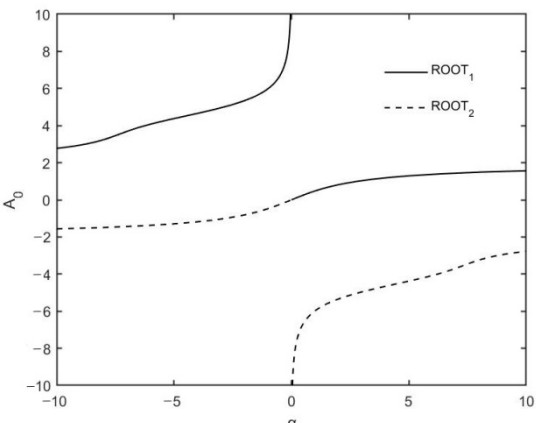

**Figure 1.** The solutions of $A_0$ vs. parameter $\alpha$.

As shown in Figure 1, there are two real solutions of $A_0$ with $\alpha$ from the nonlinear algebraic equation in Equation (17). The amplitude $A_0$ can be given with $\alpha$, but the standard Maclaurin expansion only offers expression with a relatively small convergent radius. To obtain a power series solution with relatively larger convergent radius, an auxiliary expansion scheme with Chebyshev polynomial can be utilized. The procedure starts with the expression of $A_0(\alpha)$, and the expansion is

$$A_0(\alpha) = \sum_{n=0}^\infty C_n T_n\left(\frac{\alpha}{5} - 1\right),$$
$$C_n = \frac{2}{5\pi} \int_0^{10} A_0(\alpha) \frac{T_n\left(\frac{\alpha}{5}-1\right)}{\sqrt{1-\left(\frac{\alpha}{5}-1\right)^2}} d\alpha, \tag{18}$$

where $T_n$ is the $n$th-order Chebyshev polynomial, and $C_n$ is the coefficient of expansion. In case an explicit expression of $A_0(\alpha)$ is not available, the symbolic expression from mathematical tools such as Mathematica® and MATLAB® can be utilized for the evaluation in Equation (18). Previous studies have shown that Equations (17) and (18) here have good stability and convergence [15,32–34]. In this study, from Equations (17) and (18), the explicit expression of $A_0(\alpha)$ with approximation is

$$
\begin{aligned}
A_0 = \ & -1.12113113903689 \times 10^{-7} + 0.5160330121688547\alpha \\
& -0.00009479584375745703\alpha^2 - 0.045486335180154686\alpha^3 \\
& +0.000012757993686000196\alpha^4 + 0.006466636289057949\alpha^5 \\
& +0.006958737479441875\alpha^6 - 0.012229157808245239\alpha^7 \\
& +0.008664382051570434\alpha^8 - 0.0039340381580180665\alpha^9 \\
& +0.0012851620189843023\alpha^{10} - 0.00031666548778055986\alpha^{11} \\
& +0.00006014045641459205\alpha^{12} \\
& -0.0000088771605033395703\alpha^{13} \\
& +0.0000010175292935304744\alpha^{14} \\
& -8.973639340531093 \times 10^{-8}\alpha^{15} \\
& +5.973823554191147 \times 10^{-9}\alpha^{16} \\
& -2.903563905834224 \times 10^{-10}\alpha^{17} \\
& +9.721628341667188 \times 10^{-12}\alpha^{18} \\
& -2.004035813851492 \times 10^{-13}\alpha^{19} \\
& +1.916801557948531 \times 10^{-15}\alpha^{20}
\end{aligned}
\tag{19}
$$

Then, from Equation (10) with the known value of $A_0$, the approximate solution is modified to

$$
\theta = A_0 \sin\frac{\pi}{2}x + A_1 \sin\frac{3\pi}{2}x,
\tag{20}
$$

where $A_1$ is solved. Following the same procedure as demonstrated for $A_0$, the nonlinear algebraic equation for amplitude $A_1$ is

$$
\begin{aligned}
\frac{2\alpha}{3\pi} - \frac{9\pi^2 A_1}{8} + \frac{1}{654729075\pi} & 2\alpha[230945A_0^2(189 - 27A_0^2 + A_0^4) \\
& -25194A_0(13365 - 1430A_0^2 + 59A_0^4)A_1 \\
& +14535(-5005 - 1118A_0^2 + 161A_0^4)A_1^2 \\
& -6460A_0(-6561 + 695A_0^2)A_1^3 + 285(17017 \\
& +3947A_0^2)A_1^4 - 1771470A_0A_1^5 - 138567A_1^6] = 0.
\end{aligned}
\tag{21}
$$

Then, from Equation (21) with an approximate procedure for an explicit expression, the convergent solution is

$$
A_1 = \frac{\frac{2\alpha}{3\pi} + \frac{2\alpha A_0^2(189 - 27A_0^2 + A_0^4)}{2835\pi}}{\frac{9\pi^2}{8} + \frac{4\alpha A_0(13365 - 1430A_0^2 + 59A_0^4)}{51975\pi}},
\tag{22}
$$

noting the next approximate solution is

$$
\theta = A_0 \sin\frac{\pi}{2}x + A_1 \sin\frac{3\pi}{2}x + A_2 \sin\frac{5\pi}{2}x,
\tag{23}
$$

and the algebraic equation for $A_2$ is

$$
\begin{aligned}
&104187267600\alpha A_0^6 + 403446387600\alpha A_1^6 \\
&-3205762080\alpha A_0^5(535A_1 - 751A_2) + 193391815584\alpha A_1^5 A_2 \\
&+2042887600\alpha A_1^4(-6561 + 2437A_2^2) \\
&+93000000A_1^3 A_2(-124729 + 11855A_2^2) \\
&+24812400\alpha A_0^4(-46189 + 132753A_1^2 - 206834A_1 A_2 + 17313A_2^2) \\
&+1708100000\alpha A_1 A_2(116127 - 14850A_2^2 + 625A_2^4) \\
&+35377680\alpha A_1^2(5009445 - 1890690A_2^2 + 131671A_2^4) \\
&+8595449577(-25200\alpha + 196875\pi^3 A_2 \\
&\qquad +8400\alpha A_2^2 - 560\alpha A_2^4 + 16\alpha A_2^6) \\
&-4315200\alpha A_0^3[1099929A_1^3 - 3598787A_1^2 A_2 \\
&-625A_2(-22287 + 2785A_2^2) + 23A_1(-365313 + 137021A_2^2)] \\
&+71920\alpha A_0^2[50284785A_1^4 - 115877220A_1^3 A_2 \\
&\qquad +90A_1^2(-6589523 + 2468591A_2^2) \\
&\qquad +A_1(512155260A_2 - 64424868A_2^3) \\
&\qquad -209(686205 - 254610A_2^2 + 17599A_2^4)] \\
&-160\alpha A_0[5067002235A_1^5 - 55775187135A_1^4 A_2 \\
&\qquad -41354A_1^2 A_2(-20529405 + 2545379A_2^2) \\
&\qquad +16182A_1^3(-7763305 + 2927733A_2^2) \\
&\qquad -9530625A_2(358701 - 44950A_2^2 + 1875A_2^4) \\
&\qquad +3875A_1(277272567 - 103551750A_2^2 + 7178125A_2^4)] = 0,
\end{aligned}
\tag{24}
$$

with the explicit expression of $A_2$ as

$$
\begin{aligned}
A_2 = \ & -(-216605329340400\alpha - 10314539492400\alpha A_0^2 \\
& -1146059943600\alpha A_0^4 + 104187267600\alpha A_0^6 \\
& -171908991540000\alpha A_0 A_1 + 36257169124800\alpha A_0^3 A_1 \\
& -1715082712800\alpha A_0^5 A_1 + 177222542187600\alpha A_1^2 - \\
& +42652664474400\alpha A_0^2 A_1^2 + 3293920537200\alpha A_0^4 A_1^2 \\
& +20100128241600\alpha A_0 A_1^3 - 4746413620800\alpha A_0^3 A_1^3 \\
& -13403385543600\alpha A_1^4 + 3616481737200\alpha A_0^2 A_1^4 \\
& -810720357600\alpha A_0 A_1^5 \\
& +403446387600\alpha A_1^6)/(1692229135471875\pi^3 \\
& +546983154900000\alpha A_0 - 60108039000000\alpha A_0^3 \\
& +2407527322080\alpha A_0^5 + 198356528700000\alpha A_1 \\
& +36834206299200\alpha A_0^2 A_1 - 5132047941600\alpha A_0^4 A_1 \\
& -135835682299200\alpha A_0 A_1^2 + 15529485662400\alpha A_0^3 A_1^2 \\
& -11599797000000\alpha A_1^3 - 8333889662400\alpha A_0^2 A_1^3 \\
& +8924029941600\alpha A_0 A_1^4 + 193391815584\alpha A_1^5).
\end{aligned}
\tag{25}
$$

Now with all three coefficients known, the approximate solution is given as in Equation (23), which is quite different from the power series solution produced by other methods [15].

To verify this solution and make comparisons with other known results, the rotation angle at the free end is obtained by setting $x = 1$ in Equation (24) as

$$
\theta_B = A_0 - A_1 + A_2.
\tag{26}
$$

For comparison, the exact solution from Equation (9) is [15]

$$
\sqrt{\alpha} = K(\mu^2) - F(\phi, \mu), \quad \mu = \sqrt{\frac{1 + \sin\theta_B}{2}}, \quad \phi = \arcsin\left(\frac{1}{\sqrt{2}\mu}\right).
\tag{27}
$$

Here, $K(\mu)$ is the complete elliptic function of the first kind, and $F(\phi, \mu)$ is the elliptic function of the second kind. Then, the relationship between $\alpha$ and $\theta_B$ can be analytically obtained from these equations.

The optimal approximate solution by the HAM is given as [15]

$$\theta_B = \frac{\alpha}{2}\frac{f(\alpha)}{g(\alpha)},\tag{28}$$

with

$$
\begin{aligned}
f(\alpha) = \quad & 1 + 3.98575 \times 10^{-2}\alpha^2 - 5.41174 \times 10^{-2}\alpha^4 \\
& + 5.72575 \times 10^{-3}\alpha^6 + 3.79533 \times 10^{-4}\alpha^8 \\
& - 1.51429 \times 10^{-11}\alpha^{14} - 2.29142 \times 10^{-15}\alpha^{16} \\
& - 8.87896 \times 10^{-6}\alpha^{10} + 2.63041 \times 10^{-8}\alpha^{12} \\
& - 3.45006 \times 10^{-21}\alpha^{18} - 7.00678 \times 10^{-28}\alpha^{20}, \\
g(\alpha) = \quad & 1 + 0.131524\alpha^2 - 5.99231 \times 10^{-2}\alpha^4 \\
& + 2.34466 \times 10^{-3}\alpha^6 + 9.90299 \times 10^{-4}\alpha^8 \\
& - 1.37001 \times 10^{-5}\alpha^{10} + 3.44172 \times 10^{-8}\alpha^{12} \\
& - 1.45098 \times 10^{-11}\alpha^{14} - 2.26721 \times 10^{-15}\alpha^{16} \\
& - 3.50731 \times 10^{-21}\alpha^{18} - 7.00678 \times 10^{-28}\alpha^{20}.
\end{aligned}
\tag{29}
$$

To visualize the solutions and make comparisons, different approximate solutions and the exact solution are plotted in Figure 2 with parameter $\alpha$.

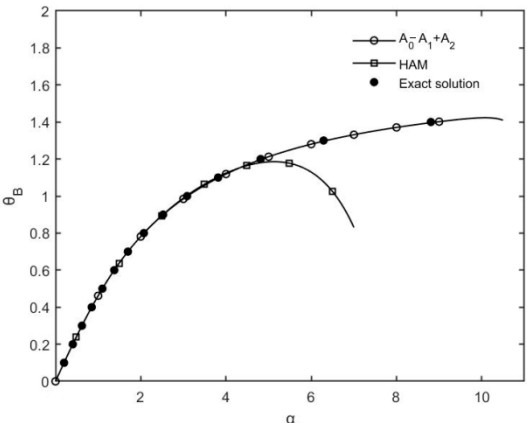

**Figure 2.** The rotation angle at the free end $\theta_B$ vs. $\alpha$ with different solutions.

It is clear from Figure 2 that the approximations are very good with just two terms. With three terms, it is almost the exact solution in the range of parameter $\alpha$, as shown in the plot. The excellent accuracy is a little surprising given the fact the procedure and technique are relatively simple, as shown in this paper. As for a comparison with the rigorous and sophisticated HAM [15], it also shows that for this particular example, this method provides a relatively accurate approximation with a much simpler procedure. We can plot the rotation angle at the free end $\theta_B$ vs. $\alpha$ with solutions of different orders in Figure 3.

From Figure 3, it can be found that there is still a noticeable gap between the first-order and the exact solutions. On the other hand, the second-order solution gradually approaches to the exact solution. The third-order solution is almost exact in a large interval in comparison with the closed-form solution with elliptic functions. It is evident now that the Galerkin method is extremely efficient in obtaining an accurate solution to the nonlinear differential equation of the elastica. The excellent choice of the trigonometric sine function as the basis of trial solutions makes the procedure ideal, and the accuracy is also surprisingly good with just the third-order solution. It is evident that the Galerkin method can be equally powerful for nonlinear problems arising from applications in many fields. The elastica problem has a long and rich history, but it is still a hot topic related to the flexure and buckling of beams and rods; accurate results are essentially obtained from the elliptic functions [38,39]. The procedure and solution from this study show that with the Galerkin method, a new technique to solve similar nonlinear differential equations is

available, and it can be much simpler to obtain reasonably accurate solutions. This was the objective of this study: to demonstrate the usefulness and efficiency of the Galerkin method with nonlinear differential equations. It is a neglected technique worth being promoted, as demonstrated by the popular example of the elastica problem.

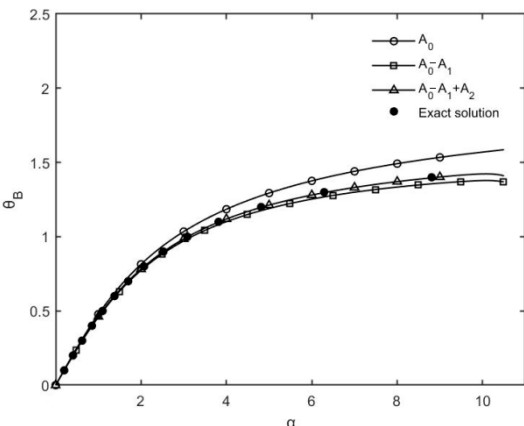

**Figure 3.** The rotation angle at the free end $\theta_B$ vs. $\alpha$ with solutions of different orders.

The choice of trial functions is of great importance in the simplification of calculation and determination of accuracy, but the choice should be made from experience in dealing with approximate solutions for the Galerkin and Rayleigh–Ritz methods, which are equivalent in most cases. Furthermore, the solution of coupled nonlinear equations of amplitudes can be hard to acquire, but the utilization of symbolic computing tools such as Mathematica® and MATLAB® can provide much better solutions with explicit expressions that can only be given and meaningful for the lower orders. For practical problems such as the elastica problem in this study, the combination of ideal trial functions, optimal procedure, iteration, and symbolic calculation produces sufficiently accurate results to meet application requirements. If a systematic procedure with symbolic computing is developed from the approximate solutions of nonlinear algebraic equations, the efficiency and accuracy can be greatly increased to achieve better solutions of nonlinear differential equations. Further studies of and improvement in the Galerkin method will add powerful tools to the collection of techniques for nonlinear differential equations. The power and capabilities of symbolic computation can make the Galerkin method favorable for some nonlinear differential problems with less numerical computation, as shown with the elastica problem in this study.

## 4. Conclusions

By utilizing the Galerkin method with a proper choice of trial functions of the approximate solution, a novel procedure to obtain asymptotic solutions of nonlinear differential equations was presented and validated. The method and procedure were verified by solving a beam flexure problem with an enhanced solution in comparison with the solution from the homotopy analysis method, with satisfaction. The procedure is straightforward for mathematical implementation because the calculation is simple, but the optimal choice of the trial function is critical due to the simplification of calculation with the orthogonality through integration. The further refinement of coefficients can be achieved with iterative procedure, which is also easy to implement in the calculation procedure. The accurate results in such a simple procedure demonstrated that similar nonlinear problems can be solved with this new and efficient method, achieving relatively accurate solutions with a limited number of terms. While the nonlinear problems are receiving attention for important applications in many natural phenomena and engineering problems, innovative methods for solving nonlinear differential equations including this Galerkin method should also be examined and refined to enrich the toolbox for addressing greater challenges posed

by analytical requirements. The use of the traditional Galerkin method for approximate solutions to nonlinear differential equations in this study is only an example to show the advantage of a simple computational procedure for a seemingly difficult problem with exact solutions in elliptic functions and a large number of numerical computation processes.

**Author Contributions:** Conception, revision, and submission of the paper, J.W.; mathematical formulation, procedure, derivation, calculation, and check, J.Z. and R.W., with help from L.W.; reviews and discussions of the study and drafting were also completed with the participation of T.M. All authors have read and agreed to the published version of the manuscript.

**Funding:** This research was supported by the National Natural Science Foundation of China (Grants 11672142, 11772163, and 12172183) with additional support from the Technology Innovation 2025 Program (Grant 2019B10122) of the Municipality of Ningbo, Zhejiang Province, China.

**Institutional Review Board Statement:** Not applicable.

**Informed Consent Statement:** Not applicable.

**Data Availability Statement:** The data from this study are available from the authors and their website.

**Conflicts of Interest:** The funders had no role in the design of the study; in the collection, analyses, or interpretation of data; in the writing of the manuscript; or in the decision to publish the results.

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
