# Peer review of "The Approximate Solution of Nonlinear Flexure of a Cantilever Beam with the Galerkin Method"

_applsci, doi:10.3390/app12136720_

Round 1

Reviewer 1 Report

The manuscript presents an approximate solution, based on the Galerkin Method, for the bending of a cantilever beam subjected to a concentrated load at its free end. the authors consider geometric nonlinearity (large deformations), taking an isotropic and elastic material and using the classical beam theory.

There are many articles published that show interesting results for the solutions of nonlinear equations using the Galerkin Method. Although the procedure proposed in this work is interesting, it is not evident what the original contribution or the highlights of the article are.

The "Introduction" is very generic, the authors mention articles that do work in a nonlinear regime, but that direct their studies to composite materials, shear deformation theories, higher order theories, etc. This makes it difficult to quantify the relationship between what other authors have already done and the hypothetical contribution of this work. On the other hand, the description made of the references is not directly related to what the authors state in the description (for example, References 2, 8, 10, etc).

The way in which the work is presented suggests that it is nothing more than a numerical exercise, rather than a formulation that contains an original contribution.

For the reasons mentioned above, I suggest you consider the following aspects:

1) Rewrite the Introduction, considering specific works on the subject and, as a conclusion to their analysis, clearly establish the original contribution of this approach.

2) Reorder some aspects of the manuscript structure. It would be easier for readers to have a diagram summarizing the computational scheme, as well as appending the results for the coefficients of the test functions.

The highlights must be clear and explicit as innovative contributions of this work.

Author Response

First of all, we thank the reviewers and editor for their support of this study.  We thank the editor and reviewers for your careful reading and thoughtful suggestions on the revision and improvement. The authors have read the comments carefully and made revisions in the consideration of the suggestions. The revisions have been highlighted in the manuscript with additional replies in this file for better explanation.

Reviewer #1

The manuscript presents an approximate solution, based on the Galerkin Method, for the bending of a cantilever beam subjected to a concentrated load at its free end. The authors consider geometric nonlinearity (large deformations), taking an isotropic and elastic material and using the classical beam theory.

There are many articles published that show interesting results for the solutions of nonlinear equations using the Galerkin Method. Although the procedure proposed in this work is interesting, it is not evident what the original contribution or the highlights of the article are.

The "Introduction" is very generic, the authors mention articles that do work in a nonlinear regime, but that direct their studies to composite materials, shear deformation theories, higher order theories, etc. This makes it difficult to quantify the relationship between what other authors have already done and the hypothetical contribution of this work. On the other hand, the description made of the references is not directly related to what the authors state in the description (for example, References 2, 8, 10, etc).

The way in which the work is presented suggests that it is nothing more than a numerical exercise, rather than a formulation that contains an original contribution.

Reply: The reviewer is definitely correct that the introduction of the paper is not focused on the nonlinear analysis of beams, rather than a general description of nonlinear problems in beams of many applications.  It is because the paper places more emphasis on the method other than on the problem.  Of course, the reviewer provided the right suggestion for improvement.  Some introductions and discussions about the elastica problem are added to enhance the focus on the applications to nonlinear elastic with larger deformation, or the elastica problem.

Q1: Rewrite the Introduction, considering specific works on the subject and, as a conclusion to their analysis, clearly establish the original contribution of this approach.

A1: Thanks.  We have revised our Introduction section per your suggestion for a better understanding of the specific application we are dealing with.

Q2: Reorder some aspects of the manuscript structure. It would be easier for readers to have a diagram summarizing the computational scheme, as well as appending the results for the coefficients of the test functions.

A2: The suggestion is taken.  Some discussions are added to relate the calculation and results.

Reviewer 2 Report

For optimal design and accruate prediction of structural behavior,the nonlinear analysis of large deformation of elastic beams has been considered  which  have many applications in various engineering fields. In this study, the nonlinearequation of flexure of an elastic beam  has been solved by the Galerkin method andthe accurate solution is compared with known results from other methods. Ageneral procedure with the Galerkin method  has demonstrated for efficientsolutions of nonlinear differential equations with the potential for favoritepromotion and implementation in more applications. The over all work is good and acceptable. However some points need to be addressed like:

1. Need of this research and superiority of the method has not explained.

2. Stability and converging of method need to be updated.

3. Literature should be updated with recent work like: Green function's properties and existence theorems for nonlinear singular-delay-fractional differential equations. Discrete & Continuous Dynamical Systems-S. 2020;13(9):2475.Hyers–Ulam stability and existence criteria for coupled fractional differential equations involving p-Laplacian operator. Advances in Difference Equations. 2018 Dec;2018(1):1-6.Stability analysis of a dynamical model of tuberculosis with incomplete treatment. Advances in Difference Equations. 2020 Dec;2020(1):1-4.m. Computational and theoretical modeling of the transmission dynamics of novel COVID-19 under Mittag-Leffler power law. Alexandria Engineering Journal. 2020 Oct 1;59(5):3133-47,Stability analysis and a numerical scheme for fractional KleinGordon equations. Mathematical Methods in the Applied Sciences. 2019 Jan 30;42(2):723-32

Author Response

For optimal design and accurate prediction of structural behavior, the nonlinear analysis of large deformation of elastic beams has been considered which has many applications in various engineering fields. In this study, the nonlinear equation of flexure of an elastic beam has been solved by the Galerkin method and the accurate solution is compared with known results from other methods. A general procedure with the Galerkin method has demonstrated for efficient solutions of nonlinear differential equations with the potential for favorite promotion and implementation in more applications. The overall work is good and acceptable.

Reply: Thank you for your positive understanding of our objective and encouraging comments.  We really want to demonstrate and promote this traditional but neglected method with the assistance of symbolic computational tools.

Q1: Need of this research and superiority of the method has not explained.

A1: Thanks for the reminder.  Our objective is to promote the utilization of the Galerkin method through solving nonlinear algebraic equations to obtain nonlinear solutions to nonlinear differential equations.  We need to explain this more clearly to readers.

Q2: Stability and converging of method need to be updated.

A2: Very good question to be considered indeed.  We did not go into details about the analysis of the stability and convergence properties with this specific equation, but the convergence is checked and validated with solutions from another method.  We think this is enough for the demonstration of the method, and the stability and convergence rate will be examined in future studies.

Q3: Literature should be updated with recent work like: Green function's properties and existence theorems for nonlinear singular-delay-fractional differential equations. Discrete & Continuous Dynamical Systems-S. 2020;13(9):2475.Hyers–Ulam stability and existence criteria for coupled fractional differential equations involving p-Laplacian operator. Advances in Difference Equations. 2018 Dec;2018(1):1-6.Stability analysis of a dynamical model of tuberculosis with incomplete treatment. Advances in Difference Equations. 2020 Dec;2020(1):1-4.m. Computational and theoretical modeling of the transmission dynamics of novel COVID-19 under Mittag-Leffler power law. Alexandria Engineering Journal. 2020 Oct 1;59(5):3133-47,Stability analysis and a numerical scheme for fractional Klein‐Gordon equations. Mathematical Methods in the Applied Sciences. 2019 Jan 30;42(2):723-32.

A3: Thanks for pointing to these references on solution methods of nonlinear differential equations. We have examined and some more references have been added to the reference list.

Reviewer 3 Report

The paper is devoted to finding approximate solution for flexural behavior of a cantilever beam using Galerkin method. The manuscript is well prepared but needs further improvements before publishing. My comments are as follows:

1)       In structural engineering, elastic and in-elastic behavior are considered and interpreted as linear and nonlinear behavior, respectively. In other words, where the system behaves elasticly, it is meaningless to express the nonlinearity of the system. Given that you have solved an engineering example, be careful not to misinterpret the definitions of linearity and nonlinearity in mathematics with the definitions of linearity and nonlinearity in structural engineering. Please review this issue and resolve any ambiguities.

2)       The abstract is very short and it is necessary to mention more details of the method along with the results.

3)       In order to increase the number of readers of MDPI journals, it is better to refer to some works published by MDPI specially “applied science” journal in the introduction section. Also, the number of self-citations is high and must be reduced. Here are some recent studies from other authors that can be used instead. https://www.mdpi.com/2076-3417/11/19/9307

https://www.mdpi.com/2076-3417/12/6/3206

4)       In Equation 9, a state of loading is considered as constant load that does not change with time. If the load entered is a function of time for example P=P0 sin(wt) where w is the frequency of excitation, how the solution must be expressed? There are also different types of beams with different support conditions and input loads (constant loads as static loads and time varying loads as dynamic loads). Keep in mind that, most engineering structures are designed for withstanding of dynamic loads in addition to static loads. In order to better show the accuracy of the proposed method, it is highly suggested to solve two other examples of beams with different support conditions and input loads (at least one example of the dynamic load and determining the vibration of the system affected by this load) and compare the outcomes with exact results.

Author Response

The paper is devoted to finding approximate solution for flexural behavior of a cantilever beam using Galerkin method. The manuscript is well prepared but needs further improvements before publishing.

 Reply: Thanks for the kind approval of our efforts in this study.

Q1: In structural engineering, elastic and in-elastic behavior are considered and interpreted as linear and nonlinear behavior, respectively. In other words, where the system behaves elasticly, it is meaningless to express the nonlinearity of the system. Given that you have solved an engineering example, be careful not to misinterpret the definitions of linearity and nonlinearity in mathematics with the definitions of linearity and nonlinearity in structural engineering. Please review this issue and resolve any ambiguities.

A1: Thanks.  We checked our paper and made sure that the nonlinearity is restricted to the nonlinear differential equations per se.  We do not imply any relation to engineering problems with in-elastic behavior.

Q2:The abstract is very short and it is necessary to mention more details of the method along with the results.

A2: Thanks for the suggestion.  The abstract is revised with additional descriptions pertaining to the equation, method, and results.

Q3: In order to increase the number of readers of MDPI journals, it is better to refer to some works published by MDPI specially “applied science” journal in the introduction section. Also, the number of self-citations is high and must be reduced. Here are some recent studies from other authors that can be used instead. https://www.mdpi.com/2076-3417/11/19/9307

https://www.mdpi.com/2076-3417/12/6/3206

A3: We appreciate this suggestion. We have added more references from the MDPI journals on beams, particularly elastica.

Q4: In Equation 9, a state of loading is considered as constant load that does not change with time. If the load entered is a function of time for example P=P0 sin(wt) where w is the frequency of excitation, how the solution must be expressed? There are also different types of beams with different support conditions and input loads (constant loads as static loads and time varying loads as dynamic loads). Keep in mind that, most engineering structures are designed for withstanding of dynamic loads in addition to static loads. In order to better show the accuracy of the proposed method, it is highly suggested to solve two other examples of beams with different support conditions and input loads (at least one example of the dynamic load and determining the vibration of the system affected by this load) and compare the outcomes with exact results.

A4: Thanks for your suggestions on future research directions with the beam equation and method.  The vibrations of beams, both free and forced, can be of interest and importance with the nonlinear equations.  The authors have studied some of these problems with beams and plates [15].  It is possible to be solved with the extended Galerkin method as shown in our earlier papers [29, 31], and we shall work on some of them with novel methods and solutions.

Round 2

Reviewer 1 Report

The authors have considered the comments and suggestions made by this reviewer. Consequently, they have incorporated the suggestions and improved the manuscript, so that the current version is in a position to be approved for publication.

Author Response

We thank the reviewer for the support of this study and the publication of this paper.

Reviewer 3 Report

All comments of the reviewer have been taken into consideration and all questions have been answered. In the reviewer point of view, the paper in its current form can be published.

Author Response

We thank the support of the publication by this reviewer.